# Development and evaluation of a live birth prediction model for evaluating human blastocysts from a retrospective study

Hang Liu[1†], Zhuoran Zhang[2†], Yifan Gu[3,4†], Changsheng Dai[1], Guanqiao Shan[1], Haocong Song[1], Daniel Li[5], Wenyuan Chen[1], Ge Lin[3,4,6,7]*, Yu Sun[1,5,8,9]*

[1]Department of Mechanical Engineering, University of Toronto, Toronto, Canada; [2]School of Science and Engineering, The Chinese University of Hong Kong-Shenzhen, Shenzhen, China; [3]Institute of Reproductive and Stem Cell Engineering, School of Basic Medical Science, Central South University, Changsha, China; [4]Reproductive and Genetic Hospital of CITIC-Xiangya, Changsha, China; [5]Department of Electrical and Computer Engineering, Toronto, Canada; [6]Key Laboratory of Reproductive and Stem Cell Engineering, National Health and Family Planning Commission, Changsha, China; [7]National Engineering Research Center of Human Stem Cells, Changsha, China; [8]Institute of Biomedical Engineering, University of Toronto, Toronto, Canada; [9]Department of Computer Science, University of Toronto, Toronto, Canada

*For correspondence:
linggf@hotmail.com (GL);
sun@mie.utoronto.ca (YS)

†These authors contributed equally to this work

## Abstract

**Background:** In infertility treatment, blastocyst morphological grading is commonly used in clinical practice for blastocyst evaluation and selection, but has shown limited predictive power on live birth outcomes of blastocysts. To improve live birth prediction, a number of artificial intelligence (AI) models have been established. Most existing AI models for blastocyst evaluation only used images for live birth prediction, and the area under the receiver operating characteristic (ROC) curve (AUC) achieved by these models has plateaued at ~0.65.

**Methods:** This study proposed a multimodal blastocyst evaluation method using both blastocyst images and patient couple's clinical features (e.g., maternal age, hormone profiles, endometrium thickness, and semen quality) to predict live birth outcomes of human blastocysts. To utilize the multimodal data, we developed a new AI model consisting of a convolutional neural network (CNN) to process blastocyst images and a multilayer perceptron to process patient couple's clinical features. The data set used in this study consists of 17,580 blastocysts with known live birth outcomes, blastocyst images, and patient couple's clinical features.

**Results:** This study achieved an AUC of 0.77 for live birth prediction, which significantly outperforms related works in the literature. Sixteen out of 103 clinical features were identified to be predictors of live birth outcomes and helped improve live birth prediction. Among these features, maternal age, the day of blastocyst transfer, antral follicle count, retrieved oocyte number, and endometrium thickness measured before transfer are the top five features contributing to live birth prediction. Heatmaps showed that the CNN in the AI model mainly focuses on image regions of inner cell mass and trophectoderm (TE) for live birth prediction, and the contribution of TE-related features was greater in the CNN trained with the inclusion of patient couple's clinical features compared with the CNN trained with blastocyst images alone.

**Conclusions:** The results suggest that the inclusion of patient couple's clinical features along with blastocyst images increases live birth prediction accuracy.

**Funding:** Natural Sciences and Engineering Research Council of Canada and the Canada Research Chairs Program.

## Editor's evaluation

This article provides important findings that have practical implications for reproductive medicine and would be of interest to IVF specialists. Based on the compelling strength of evidence, the study demonstrates significant results in improving the predictive value of the live birth model based on blastocyst evaluation and clinical features.

## Introduction

Infertility is a global health issue, affecting more than 50 million couples worldwide (*Mascarenhas et al., 2012*). Since the birth of the first in vitro fertilization (IVF) child in 1978, over 8 million children were born with IVF treatment (*Adamson et al., 2018*). Among the various factors contributing to IVF outcomes, the quality of the blastocyst (day 5 embryo) selected for transfer is critical for the success of IVF treatment. Manual grading of blastocyst development stage, inner cell mass (ICM), and trophectoderm (TE) remains the most common method for blastocyst evaluation. While the blastocyst morphological grading is widely used in clinical practice, morphological grades of the development stage, ICM, and TE have shown limited predictive power on clinical outcomes (*Seli et al., 2011*; *Reignier et al., 2019*; *Bartolacci et al., 2021*; *Ueno et al., 2021*; *Xiong et al., 2022*). It is desired to identify features for accurate prediction of clinical outcomes of blastocysts.

To achieve this goal, the convolutional neural network (CNN) is expected to play a critical role. CNN is able to automatically detect discriminative features from images and has been the state-of-the-art method in various fields in medical imaging, such as lung cancer prediction (*Ardila et al., 2019*), breast cancer prediction (*McKinney et al., 2020*), and diabetic retinopathy screening (*Bora et al., 2021*). To apply CNN to predict the clinical outcome of a blastocyst, images of blastocysts with a known clinical outcome (e.g., pregnancy and live birth) are collected for the CNN model development. The area under the receiver operating characteristic (ROC) curve (AUC) is the most commonly used metric to evaluate and compare machine learning models on predicting clinical outcomes of blastocysts (*Kragh and Karstoft, 2021a*). The AUCs reported in the literature using CNN to predict clinical outcomes from blastocyst images range from 0.64 to 0.71 for pregnancy prediction (*VerMilyea et al., 2020*; *Kragh et al., 2021b*; *Berntsen et al., 2022*; *Enatsu et al., 2022*; *Loewke et al., 2022*), and are around 0.65 for live birth prediction (*Miyagi et al., 2019*; *Nagaya and Ukita, 2021*).

Besides using blastocyst images, attempts have also been made to use time-lapse videos for live birth prediction. These videos contain the entire development process from days 0 to 5–7. However, results in the literature show that CNN using static images achieved similar or slightly better accuracies in predicting clinical outcomes than using time-lapse videos, for instance, AUC=0.68–0.71 (*VerMilyea et al., 2020*; *Enatsu et al., 2022*; *Loewke et al., 2022*) versus 0.64–0.67 (*Kragh et al., 2021b*; *Berntsen et al., 2022*) for pregnancy prediction, and 0.66 (*Miyagi et al., 2019*) versus 0.65 (*Nagaya and Ukita, 2021*) for live birth prediction. A potential reason is that the redundant frames of images in time-lapse videos may work as noise causing the model to overfit and thus leading to a lower prediction accuracy (*Zhu et al., 2018*; *Wu et al., 2021*; *Tao et al., 2022*). Therefore, we opted to use static blastocyst images in this study.

Different from using blastocyst images alone to predict clinical outcomes, *Miyagi et al., 2020* proposed to use blastocyst images together with maternal clinical features including maternal age, AMH, and BMI and reported an AUC of 0.74, the highest accuracy in literature (*Miyagi et al., 2020*). However, two questions remain elusive. First, the contribution of blastocyst images and the additional contribution of maternal clinical features to live birth prediction are unknown. Second, endometrium status-related features, such as endometrium thickness and pattern, are also critical factors impacting live birth outcomes (*Ng et al., 2007*; *Bu et al., 2016*; *Mahutte et al., 2022*), but were not considered.

In this study, we quantified the effect of blastocyst images and the combined effect of both blastocyst images and patient couple's clinical features on live birth prediction. The live birth prediction model using only blastocyst images achieved an AUC of 0.67, which was significantly outperformed by the AUC of 0.77 achieved by the model using both blastocyst images and patient couple's clinical

**eLife digest** More than 50 million couples worldwide experience infertility. The most common treatment is in vitro fertilization (IVF). Fertility specialists collect eggs and sperm from the prospective parents. They combine the egg and sperm in a laboratory and allow the fertilized eggs to develop for five days into a multi-celled blastocyst. Then, the specialists select the healthiest blastocysts and return them to the patient's uterus.

Since 1978, more than 8 million children have been conceived through IVF. Yet, only about 30% of IVF attempts result in a successful birth. As a result, fertility patients often undergo multiple rounds of IVF, which can be expensive and emotionally draining. Several factors determine IVF success, one of which is the health of the blastocysts selected for transfer to the uterus. Specialists select the blastocysts using several criteria. But these human assessments are subjective and inconsistent in predicting which ones are most likely to result in a successful birth. Recent studies suggest artificial intelligence technology may help select blastocysts.

Liu et al. show that using artificial intelligence to assess blastocysts and fertility patient characteristics leads to more accurate predictions about which blastocysts are likely to result in a successful birth. In the experiments, the researchers trained an artificial intelligence computer program using pictures of 17,580 blastocysts with known birth outcomes and the parents' clinical characteristics. The model identified 16 parental factors associated with birth outcomes. The top 5 most predictive parental factors were maternal age, the day of blastocyst transfer to the uterus, how many eggs were present in the ovaries, the number of eggs retrieved and the thickness of the uterus lining. The program achieved the highest prediction of healthy births so far, compared to success rates listed in other studies.

Artificial intelligence-aided blastocyte selection using patient and blastocyst characteristics may improve IVF success rates and reduce the number of treatment cycles patient couples undergo. Before specialists can use artificial intelligence in their clinics, they must conduct confirmatory clinical studies that enroll patient couples to compare conventional methods and artificial intelligence.

features (p value<0.0001). Additionally, when endometrium status-related features (e.g., endometrium thickness and pattern) were excluded, the AUC of the model using both blastocyst images and patient couple's clinical features significantly decreased to 0.74 (p value<0.0001), indicating that the inclusion of endometrium status-related clinical features helps improve live birth prediction accuracy. Sixteen patient couple's clinical features were identified to be most related to live birth outcomes of blastocysts, among which maternal age, the day of blastocyst transfer, antral follicle count (AFC), retrieved oocyte number, and endometrium thickness measured before transfer are the top five features contributing to live birth prediction. Additionally, the CNN heatmaps showed that the CNN mainly focused on ICM and TE for live birth prediction, and the contribution of TE-related features was greater in the CNN trained with the inclusion of patient couple's clinical features compared with the CNN trained with blastocyst images alone.

## Methods
### Data set collection

We used retrospectively collected data to develop the live birth prediction model. Transferred blastocysts with known live birth outcomes for patients who underwent frozen embryo transfer cycles from 2016 to 2020, at the Reproductive and Genetic Hospital of CITIC-Xiangya, were reviewed for inclusion in the data set. Informed consent was not necessary because this study used retrospective and fully de-identified data, no medical intervention was performed on the subject, and no biological samples from the patient were collected. This study was approved by the Ethics Committee of the Reproductive and Genetic Hospital of CITIC-Xiangya (approval number: LL-SC-2021-008).

Blastocyst images were captured before transfer using a standard optical light microscope mounted with a camera. Two grayscale images were captured for each blastocyst, one focusing on ICM and the other focusing on TE. Blastocysts were cropped from the original images which have a resolution of 1024×768 and were consistently padded to 500×500 to facilitate model training. Patient

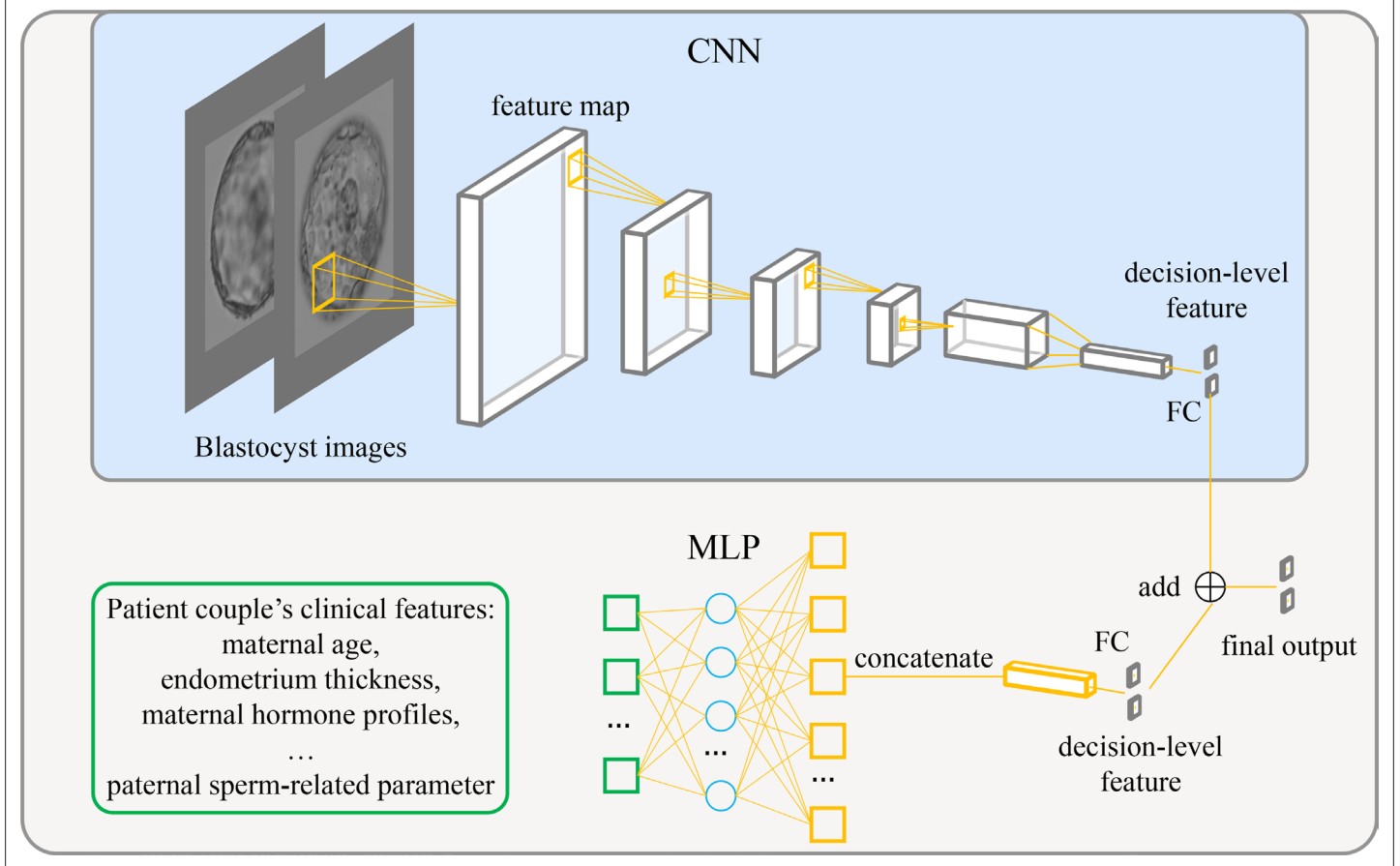

**Figure 1.** Architecture of the live birth prediction model based on multimodal blastocyst evaluation. CNN, convolutional neural network; FC, fully connected layer; MLP, multilayer perceptron.

The online version of this article includes the following source data for figure 1:

**Source data 1.** Source code to reproduce the model.

couple's clinical features consist of 103 features including maternal age and BMI, the day of blastocyst transfer, infertility diagnosis and treatment history of patient couples, ovarian stimulation protocols, maternal hormone profiles, and ultrasound results measured during the ovarian stimulation process and before transfer, and paternal semen diagnosis results (see *Supplementary file 1* for a complete list). Based on p value analysis and logistic regression (LR)-based sequential forward feature selection (*Solorio-Fernández et al., 2020*; *Raschka, 2018*), 16 clinical features that are most relevant to live birth prediction were identified and used for training the machine learning model (see Figure 3). Feature selection reduces the input feature dimensions by removing redundant features and features with limited predictive power, thus improving the model generalization capability (see *Figure 3— figure supplement 1*). The LR-based feature selection was used due to its computing efficiency, we also presented the result of multilayer perceptron (MLP)-based feature selection in *Figure 3—figure supplement 1* and *Figure 3—figure supplement 2*.

A total of 28,118 blastocysts with known live birth outcomes were reviewed, among which 17,580 blastocysts with two blastocyst images and all the 16 clinical features available were included in the data set.

## Model architecture

*Figure 1* shows the architecture of the live birth prediction model based on multimodal blastocyst evaluation. It consists of a CNN to process blastocyst images and an MLP to process patient couple's clinical features. Features from the CNN and the MLP are fused; thus, the model can be trained to simultaneously take into account both blastocyst images and patient couple's clinical features for live

birth prediction. The last fully connected layer in the CNN and the last fully connected layer in the MLP each output a decision-level feature, which has two variables used to classify the blastocyst into the positive or negative live birth outcome category. The adding operation fuses decision-level features from the CNN and the MLP, and the result of addition is taken as the final output of the overall live birth prediction model.

## Model implementation and training

The proposed live birth prediction model used EfficientNetV2-S as the backbone CNN. EfficientNetV2-S is the baseline model in the EfficientNetV2 family, which is a new family of CNN models that provide higher accuracy and training speed than conventional models (*Tan and Le, 2021*). In our work, the output dimension of the final fully connected layer in EfficientNetV2-S was set to be two, representing the positive and negative live birth outcome of a blastocyst, respectively. The model was implemented using PyTorch 1.10.1 (*Paszke et al., 2019*).

Each of the 17,580 blastocysts had two images taken at different focal planes, one focused more on TE cells and the other on ICM. Furthermore, for each blastocyst, live birth outcomes and all the 16 patient couple's clinical features were available. The blastocysts were randomly split into 80%:10%:10% to construct the training, validation, and testing data sets. The stratified random sampling approach was used to ensure that all split data sets have the same distribution of minority and majority classes. Since the ratio of blastocysts with a positive live birth outcome in the data set is 0.368, to mitigate the model's prediction bias toward the majority category (i.e., the negative live birth outcome), the weighted sampling approach, which can help rebalance the class distributions when sampling from an imbalanced data set (*Feng et al., 2021*), was employed for training the model. In the weighted sampling approach, the probability of each item to be selected is determined by its weight, and the weight of each item is assigned by inverse class frequencies. In this way, the weighted sampling approach rebalances the class distributions by oversampling the minority class and under-sampling the majority class. We also verified the approach of using weighted cross-entropy loss, which assigns greater weights to the loss caused by the prediction error of minority classes. Both approaches helped mitigate the prediction bias toward the majority class, and the results showed that the weighted sampling approach outperformed the weighted cross-entropy loss method.

Model performance is subject to training hyperparameters (e.g., optimizer, learning rate, batch size, and number of layers). Hence, an automatic hyperparameter-tuning tool is used, Facebook Ax (version 0.2.2, https://github.com/facebook/Ax), to search for the optimal hyperparameters for model training. The selected hyperparameters for training the model include a batch size of 16, an SGD optimizer with a learning rate of 0.008, and a momentum of 0.39, and three hidden layers in the MLP. A dropout layer follows each hidden layer in the MLP to prevent overfitting. The number of nodes in each hidden layer is 6836, 5657, and 468, respectively. The dropout rate in each dropout layer is 0.01, 0.07, and 0.67, respectively. The model was trained with four RTX A6000 GPUs. It took about 30 hr to search for the optimal hyperparameters and about an hour to train the model using the optimal hyperparameters.

## Statistical analysis

Statistical tests were calculated to compare clinical features between blastocysts with the positive live birth outcome and blastocysts with the negative live birth outcome. Chi-squared test was used for categorical features, t test was used for numerical features. Chi-squared test and t test were performed using Python (version 3.6). ROC curves were compared by the DeLong test implemented in MedCalc software (version 20). All statistical tests were two-tailed and considered significant if p value≤0.05.

## Results

### The inclusion of patient couple's clinical features increased AUC for live birth prediction

To quantify the individual effect of blastocyst images and the combined effect of patient couple's clinical features, we built and compared models that (1) used only blastocyst images for live birth prediction, and (2) used both blastocyst images and patient couple's clinical features for prediction.

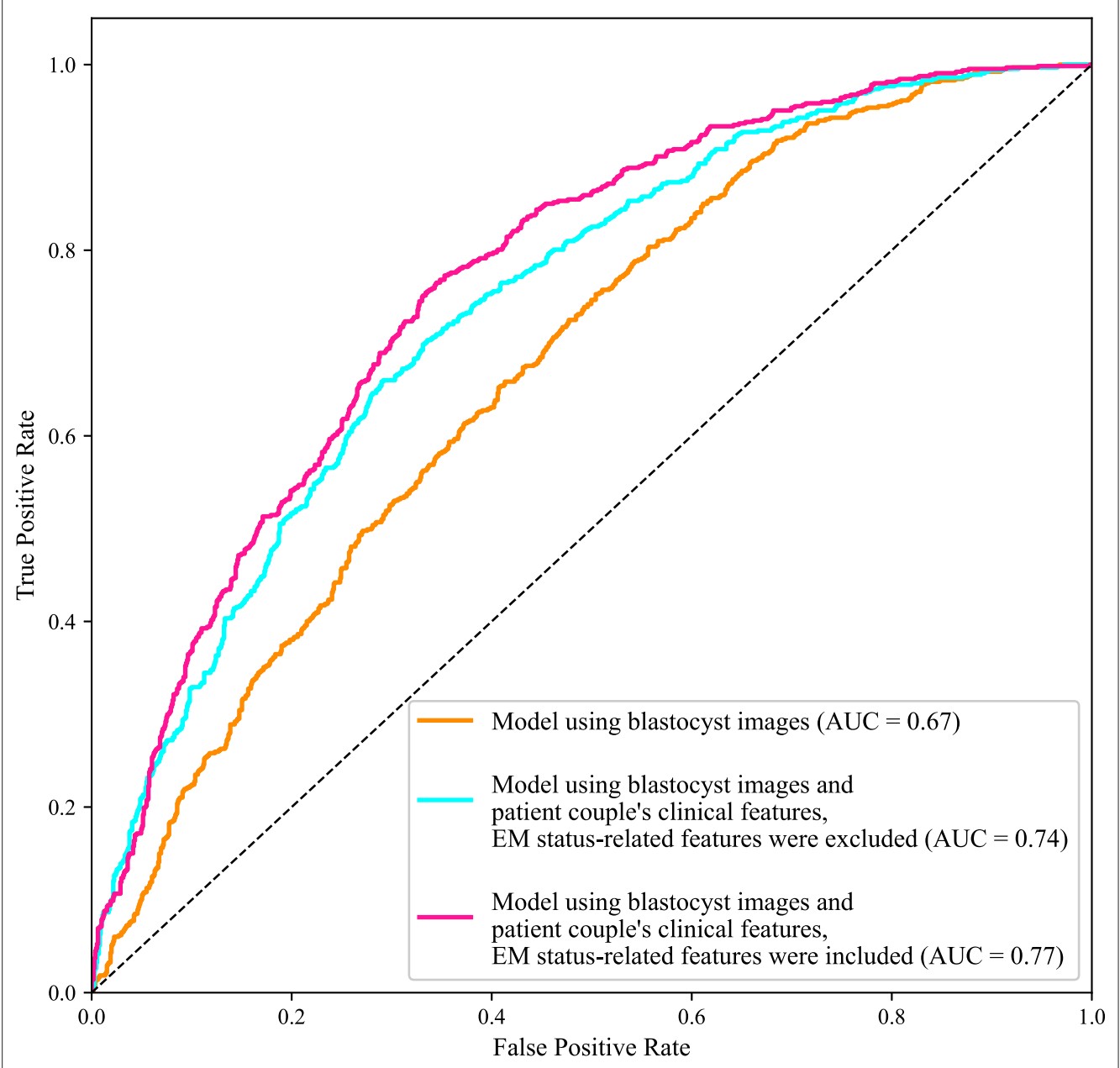

**Figure 2.** Receiver operating characteristic (ROC) analysis. ROC curves of the model using only blastocyst images, the model using blastocyst images and patient couple's clinical features where EM-status related features were excluded, and the model using blastocyst images and patient couple's clinical features where EM-status related features were included to predict live birth outcomes of 1758 blastocysts in the test data set. AUC, area under the ROC curve; EM, endometrium; EM status-related features, endometrium thickness before transfer, endometrium thickness on HCG day, endometrium pattern B (yes/no) on HCG day.

The online version of this article includes the following source data for figure 2:

**Source data 1.** Code and data used to generate the ROC curves.

In addition, to quantify the specific effect of endometrium status-related features (i.e., endometrium thickness before transfer, endometrium thickness on HCG day, and endometrium pattern B (yes/no) on HCG day) on live birth prediction, a third model trained using blastocyst images and patient couple's clinical features where endometrium status-related features were excluded, was also built and compared.

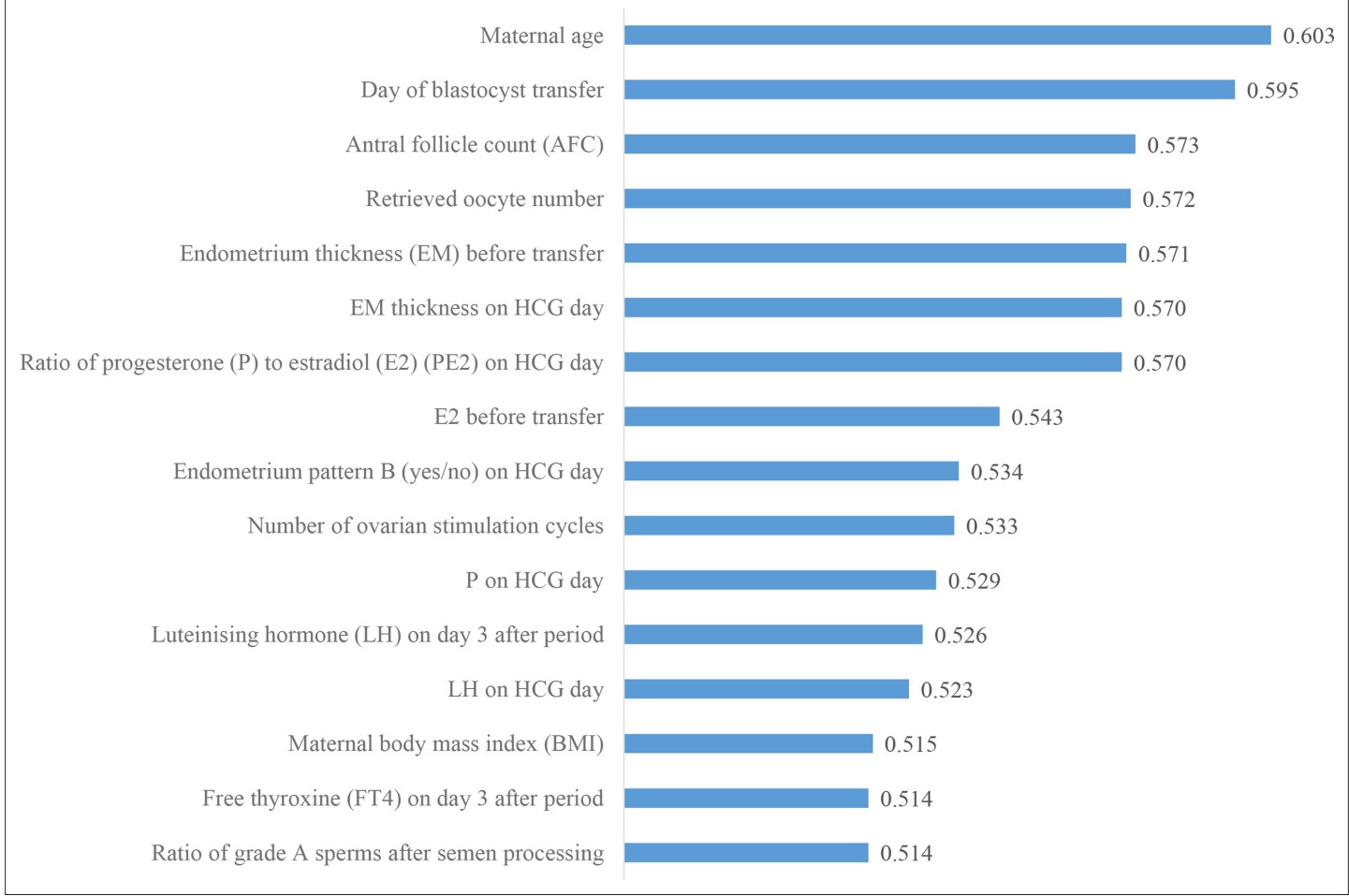

**Figure 3.** Ranking the predictive power of patient couple's clinical features. The 16 patient couple's clinical features that were identified to be most related to the live birth outcomes of the blastocysts ranked by the AUC for individually predicting live birth outcome. AUC, area under the curve.

The online version of this article includes the following source data and figure supplement(s) for figure 3:

**Source data 1.** Code and data used to generate the AUC ranking chart.

**Figure supplement 1.** ROC curves of the model using LR-selected clinical features and blastocyst images, the model using MLP-selected clinical features and blastocyst images, and the model using all clinical features showing statistical significance and blastocyst images.

**Figure supplement 2.** Ranking the predictive power of multilayer perceptron (MLP)-selected patient couple's clinical features.

*Figure 2* shows the ROC curves and AUCs of the three models for predicting live birth outcomes of 1758 blastocysts (i.e., 10% of 17,580) in the test data set. Using only blastocyst images for live birth prediction gave an AUC of 0.67, with a 95% confidence interval (CI) of 0.65–0.70. Using blastocyst images and patient couple's clinical features (endometrium status-related features excluded) significantly increased the AUC to 0.74 (95% CI: 0.72–0.76, p value<0.0001). Using both blastocyst images and patient couple's clinical features (endometrium status-related features included) achieved a prediction AUC of 0.77 (95% CI: 0.75–0.79), which is significantly higher than using only blastocyst images for prediction (p value<0.0001) and than using blastocyst images and patient couple's clinical features where endometrium status-related features were excluded (p value=0.007).

## Ranking the predictive power of patient couple's clinical features

We then investigated the predictive power of each patient couple's clinical feature in predicting live birth outcome. *Figure 3* shows the 16 features that were identified to be most related to the live birth outcomes of the blastocysts. These features were ranked according to their AUCs for individually predicting the live birth outcomes of blastocysts using univariable LR. The AUC for each feature was reported as the mean AUC over a tenfold cross-validation process.

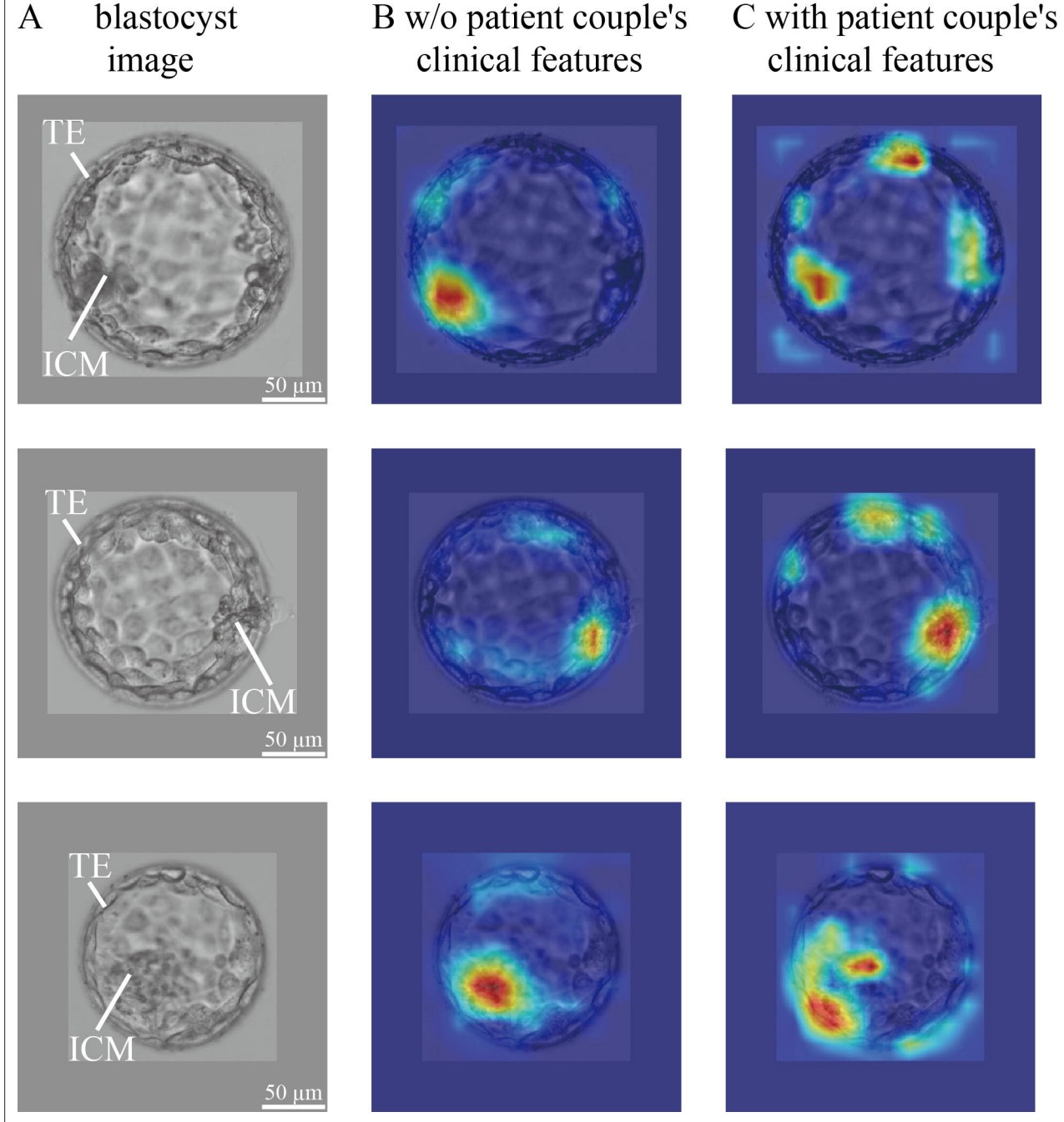

**Figure 4.** CNN heatmaps analysis. Heatmaps of the CNN trained without and with patient couple's clinical features. Column (**A**): original blastocyst images. Column (**B**): corresponding heatmaps of the CNN trained without including patient couple's clinical features. Column (**C**): corresponding heatmaps of the CNN trained with the inclusion of patient couple's clinical features. CNN, convolutional neural network.

The online version of this article includes the following source data, source code, and figure supplement(s) for figure 4:

**Source data 1.** Code and data used to generate the heatmaps shown in *Figure 4*.

**Figure supplement 1.** Two-channel blastocyst images, one channel focusing on the ICM, the other channel focusing on the TE of a same blastocyst.

**Figure supplement 1—source code 1.** Code for *Figure 4—figure supplement 1*.

## CNN heatmaps

In blastocyst images, what does CNN focus on to predict the live birth outcome of a blastocyst? Is there a difference in what the CNN focuses on between the model trained without and with the inclusion of patient couple's clinical features? To answer these questions, we used the class activation mapping method to generate heatmaps. *Figure 4* shows blastocyst images, corresponding heatmaps of the CNN trained without including patient couple's clinical features, and corresponding heatmaps of the CNN trained with including patient couple's clinical features.

The blastocyst images were cropped and padded to a consistent size to facilitate model training. The padding value was calculated as the mean pixel value of blastocyst images in the data set. Heatmaps were generated by XGradCAM (*Fu et al., 2020*). Note that the CNN takes two-channel blastocyst images as the input, one focusing on ICM and the other focusing on TE. The blastocyst images shown in *Figure 4* are those focused on ICM, and *Figure 4—figure supplement 1* shows the two-channel blastocyst images. As can be seen in *Figure 4*, when trained using only blastocyst images, the CNN mainly focuses on ICM and TE for predicting live birth outcomes. When training with both blastocyst images and patient couple's clinical features, TE-related features contributed more to live birth prediction compared with training with blastocyst images only.

## Discussion

In this study, the individual effect of blastocyst images and the combined effect of patient couple's clinical features for live birth prediction were quantified by comparing the AUC values of the model using only blastocyst images and the model using both blastocyst images and patient couple's clinical features. An AUC of 0.67 was achieved with blastocyst images only while using both blastocyst images and patient couple's clinical features led to a significantly higher AUC of 0.77 in live birth prediction. When endometrium status-related features were excluded from patient couple's clinical features, the AUC of the live birth prediction model significantly decreased (p value=0.007) from 0.77 to 0.74, indicating the strong relevance of endometrium status-related features in live birth prediction. Sixteen patient couple's clinical features were identified to be most related to live birth outcomes of blastocysts, among which maternal age, the day of blastocyst transfer, AFC, retrieved oocyte number, and endometrium thickness before transfer are the top five features contributing to live birth prediction.

This study was based on a comprehensive multimodal data set collected for blastocyst evaluation. The data set includes 17,580 blastocysts with known live birth outcomes, blastocyst images, and 16 patient couple's clinical features. As shown in *Figure 3*, 16 patient couple's clinical features comprehensively include maternal basal characteristics (age and BMI); hormone profiles measured after period (LH and FT4), on HCG day (PE2, P, and LH), and before transfer (E2); endometrium status-related features (endometrium thickness on HCG day and before transfer, endometrium pattern on HCG day); features related to oocytes (AFC, retrieved oocyte number); the day of blastocyst transfer; number of ovarian stimulation cycles; and paternal features (the ratio of grade A sperm after semen processing). For comparison, the data set studied by *Miyagi et al., 2020* did not contain endometrium status-related features and key hormone profiles (e.g., P, E2, and LH). There are numerous IVF data sets containing over 100,000 records of clinical features and live birth outcomes (*Nelson and Lawlor, 2011*; *McLernon et al., 2016*; *La Marca et al., 2021*); however, there are no blastocyst images in these data sets, and thus, these data sets cannot be used for building models to evaluate blastocysts from their images.

To handle the multimodal data set, our proposed model was designed to integrate two modules including a CNN and an MLP to enable the model to simultaneously consider images and numerical clinical features for blastocyst evaluation. The large and comprehensive multimodal data set and the proposed CNN+MLP model resulted in the highest AUC value of 0.77 ever reported thus far for predicting live birth outcomes of blastocysts. They also enabled us to quantify the predictive power of each feature in predicting the live birth outcomes of blastocysts.

The blastocyst grading system introduced in 1999 (*Gardner and Schoolcraft, 1999*; *Gardner, 1999*) remains the most common method used by embryologists to evaluate blastocyst quality although the morphological grades of blastocyst development stage, ICM and TE have limited predictive power on live birth outcomes (e.g., AUC=0.58–0.61 for live birth prediction reported by *Reignier et al., 2019*; *Bartolacci et al., 2021*; *Xiong et al., 2022*). Since CNN became a state-of-the-art

method for image-based classification, many attempts have been made to apply the CNN to blastocyst evaluation for predicting clinical outcomes (e.g., *VerMilyea et al., 2020*; *Kragh et al., 2021b*; *Berntsen et al., 2022*; *Enatsu et al., 2022*; *Loewke et al., 2022*; *Miyagi et al., 2019*; *Nagaya and Ukita, 2021*). Among these, the AUC values reported in the literature using blastocyst images only were around 0.65 for live birth prediction (*Miyagi et al., 2019*; *Nagaya and Ukita, 2021*). Similarly, we achieved an AUC of 0.67 (see *Figure 2*). Compared with the AUC of 0.58–0.61 reported in the literature using Gardner grades for live birth prediction, these results confirmed that CNN can achieve a higher prediction accuracy. As shown in *Figure 4*, the CNN mainly focuses on ICM and TE. Different from the Gardner-defined TE grade on the number of TE cells and the cohesiveness of TE cells as a whole, the CNN tends to focus on specific TE clusters. Understanding the heatmaps further requires more investigations.

*Miyagi et al., 2020* used both blastocyst images and maternal clinical features (age, AMH, and BMI) to predict live birth outcomes and achieved an AUC of 0.74 (*Miyagi et al., 2020*). The additional contribution from the three maternal clinical features was not clear since no AUC was reported by using blastocyst images alone. Furthermore, despite their importance in pregnancy and live birth, endometrium status-related features were not considered in their work. Therefore, our study used a comprehensive data set and quantitatively compared the AUC values of live birth prediction using only blastocyst images versus using both blastocyst images and patient couple's clinical features. We also quantified the usefulness of endometrium status-related features in working with blastocyst images to improve live birth prediction. Furthermore, we revealed that hormone profiles such as E2, LH, P, and FT4, features related to oocyte retrieval such as AFC and number of oocytes retrieved, and the ratio of grade A sperm after processing representing semen quality are able to work with blastocyst images to further improve the live birth prediction accuracy. Note that in this study, only the total testosterone (T) was analyzed, and free T or bioavailable T was not available for clinical feature analysis (see *Supplementary file 1*). This may cause potential bias in determining the significance of testosterone as a predictor of live birth.

Another finding of this study, by comparing the heatmaps of the CNN trained without and with the inclusion of patient couple's clinical features, is that the weights of TE-related features increased (see *Figure 4*). A potential reason may be that TE and the endometrium status-related features (e.g., endometrium thickness and pattern) play critical roles when a blastocyst initiates implantation, and a positive live birth outcome is not possible without the success of this implantation process (*Ahlström et al., 2011*; *Hill et al., 2013*; *Chen et al., 2014*; *Bakkensen et al., 2019*).

In conclusion, in this retrospective study involving 17,580 blastocysts with known live birth outcomes, blastocyst images and 16 patient couple's clinical features, we built a live birth prediction model based on multimodal blastocyst evaluation using both blastocyst images and patient couple's clinical features. We quantified the individual effect of blastocyst images and the combined effect of patient couple's clinical features on live birth prediction. Results demonstrated that using both blastocyst images and patient couple's clinical features can significantly improve live birth prediction than using blastocyst images alone.

The proposed live birth prediction model improves the evaluation of a blastocyst in terms of its live birth potential for best blastocyst selection from multiple blastocysts of a patient. The next step is to validate the model's prediction accuracy using prospectively collected data and verify its effectiveness in blastocyst selection via a randomized controlled trial (RCT). Patients enrolled in the RCT will be split into the study group and the control group (1:1 ratio). In the study group, the model selects a top blastocyst having the highest probability of live birth for transfer, and in the control group, embryologists select a top blastocyst based on their routine morphological grading for transfer. Live birth outcomes of both groups will be tracked and compared.

## Additional information

### Funding

| Funder | Grant reference number | Author |
|---|---|---|
| Natural Sciences and Engineering Research Council of Canada | | Yu Sun |
| Canada Research Chairs | | Yu Sun |

The funders had no role in study design, data collection and interpretation, or the decision to submit the work for publication.

### Author contributions

Hang Liu, Conceptualization, Data curation, Software, Formal analysis, Validation, Investigation, Visualization, Methodology, Writing – original draft, Writing – review and editing; Zhuoran Zhang, Yifan Gu, Conceptualization, Resources, Data curation, Software, Formal analysis, Validation, Investigation, Visualization, Methodology, Writing – original draft, Writing – review and editing; Changsheng Dai, Guanqiao Shan, Conceptualization, Data curation, Software, Formal analysis, Investigation, Writing – review and editing; Haocong Song, Data curation, Software, Investigation, Writing – review and editing; Daniel Li, Data curation, Software, Validation, Investigation, Writing – review and editing; Wenyuan Chen, Conceptualization, Resources, Data curation, Supervision, Validation, Investigation, Project administration, Writing – review and editing; Ge Lin, Conceptualization, Resources, Supervision, Project administration, Writing – review and editing; Yu Sun, Conceptualization, Resources, Data curation, Formal analysis, Supervision, Funding acquisition, Validation, Investigation, Visualization, Methodology, Writing – original draft, Project administration, Writing – review and editing

### Author ORCIDs

Hang Liu  http://orcid.org/0000-0001-7948-4236
Guanqiao Shan  http://orcid.org/0000-0002-2570-769X
Yu Sun  http://orcid.org/0000-0001-7895-0741

### Ethics

Human subjects: Informed consent was not necessary because this study used retrospective and fully de-identified data, no medical intervention was performed on the subject, and no biological samples from the patient were collected. This study was approved by the Ethics Committee of the Reproductive and Genetic Hospital of CITIC-Xiangya (approval number: LL-SC-2021-008).

### Decision letter and Author response

Decision letter https://doi.org/10.7554/eLife.83662.sa1
Author response https://doi.org/10.7554/eLife.83662.sa2

## Additional files

### Supplementary files
- MDAR checklist
- Supplementary file 1. P value analysis result of 103 patient couple's clinical features.

### Data availability

All processed data and code needed to reproduce the findings of the study are made openly available in deidentified form. This can be found in https://github.com/robotVisionHang/LiveBirthPrediction_Data_Code (copy archived at *Liu et al., 2023*), and attached to this manuscript. All codes and software used to analyze the data can also be accessed through the link. Due to data privacy regulations of patient data, raw data cannot be publicly shared. Interested researchers are welcome to contact the corresponding author with a concise project proposal indicating aims of using the data and how they will use the data. The project proposal will be firstly assessed by Prof. Yu Sun, Prof. Ge Lin, and

then by the Ethics Committee of the Reproductive and Genetic Hospital of CITIC-Xiangya. There are no restrictions on who can access the data.

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
