## [Editor Report]

This article provides important findings that have practical implications for reproductive medicine and would be of interest to IVF specialists. Based on the compelling strength of evidence, the study demonstrates significant results in improving the predictive value of the live birth model based on blastocyst evaluation and clinical features.

---

## [Decision Letter]

**Decision letter after peer review:**

Thank you for submitting your article "Development and evaluation of a live birth prediction model for evaluating human blastocysts: a retrospective study" for consideration by *eLife*. Your article has been reviewed by 2 peer reviewers, and the evaluation has been overseen by a Reviewing Editor and Ricardo Azziz as the Senior Editor. The reviewers have opted to remain anonymous.

Essential revisions:

1) Please, expand the section "Model architecture" and clarify the details regarding "decision-level features".

2) Provide the details of how parameter optimization was accomplished as well as the architectural details, i.e., the number of layers and nodes. What was the computational overhead for training these models?

3) Consider presenting the data for all significant predictors to justify the choice for inclusion in the model and verify if the important or top features MLP (using explanation methods) uses for prediction are the same as those inferred by the logistic regression.

4) Please, consider presenting in detail a weighted sampling approach used to tackle the imbalance issue.

5) The code is available only for generating figures 2, 4 reported in the paper. For figure 3, only data is available. Consider presenting this code for reproducibility purposes.

6) Please, improve the discussion of the potential applications of the proposed model in clinical settings and mention the method of testosterone assessment as a study limitation.

*Reviewer #1 (Recommendations for the authors):*

The authors could mention the method of testosterone assessment as a study limitation that may cause a potential bias regarding the estimation of significance of testosterone as a predictor of live birth.

The authors could consider presenting the data for all significant predictors (of example, in the supplemental Figure 3 data) and justify the choice for inclusion in the model. It will better demonstrate the correctness of the selection of predictors

*Reviewer #2 (Recommendations for the authors):*

I quite enjoyed reading the article. Overall, the motivation and concepts are well defined; however, the manuscript lacks the necessary methodological details, which hindered my ability to fully understand and appreciate the work.

The link between CNN and MLP architecture in the final integrated model is unclear. The section "Model architecture" needs to be expanded. It is not clear what "decision-level features" are from CNN or MLP. Are these features from the CNN's fully connected layer? In MLP, are they before the final layer? And how do authors concatenate these features? These details are important to understand final architecture.

How was parameter optimization accomplished? Architectural details, i.e., the number of layers and nodes, are missing. What was the computational overhead for training these models?

It is difficult to understand the discrepancy of features between CNN with clinical features and CNN without clinical features. Maybe it is because model architecture is not well defined. For the moment, it seems like CNN was trained independently, even in the concatenated version of the model. How can you explain the discrepancy between the activation maps of these two models?

Important features were identified using logistic regression. I do not observe the link presenting these features as important features when the model was built using MLP instead. Could you verify that the important or top features MLP (using explanation methods) uses for prediction are the same as those inferred by the logistic regression?

The imbalance issue was tackled using a weighted sampling approach. This approach needs to be detailed in the main text. And how were train, val, and test partitions built in view of the distribution of minority and majority classes? Did the author verify other approaches that can help resolve this issue?

The code to reproduce the model is missing.

Discussion regarding implementation in a clinical setting would be informative. How feasible is the model's deployment in a clinic? Maybe you can further elaborate on prospective clinical trial which was mentioned in line # 363.

*Reviewer #3 (Recommendations for the authors):*

The study is well designed as the Materials and methods were convincing. Results are supporting the Aims. Further studies will be important for establishing the Criteria.

The challenging issue is to keep reporting the effectiveness of this predictive procedure and publish it with LBR.

---

## [Author Response]

Essential revisions:1) Please, expand the section "Model architecture" and clarify the details regarding "decision-level features".

The decision-level features are generated from the last fully connected layer in the CNN and the last fully connected layer in the MLP. Each decision-level feature has two variables used to classify the blastocyst into the positive or negative live birth outcome category. The two decision-level features are fused by the adding operation.

To clarify this issue, we have revised the “Model architecture” section, added text in Figure 1 to indicate the last fully connected layer, and provided the code to reproduce the model. The source code has been submitted with the revised manuscript and can also be accessed at https://github.com/robotVisionHang/LiveBirthPrediction_Data_Code.

The revised “Model architecture” section now reads:

“Figure 1 shows the architecture of the live birth prediction model based on multi-modal blastocyst evaluation. It consists of a CNN to process blastocyst images and an MLP to process patient couple's clinical features. Features from the CNN and the MLP are fused; thus, the model can be trained to simultaneously take into account both blastocyst images and patient couple's clinical features for live birth prediction. The last fully connected layer in the CNN and the last fully connected layer in the MLP each output a decision-level feature, which has two variables used to classify the blastocyst into the positive or negative live birth outcome category. The adding operation fuses decision-level features from the CNN and the MLP, and the result of addition is taken as the final output of the overall live birth prediction model.”

2) Provide the details of how parameter optimization was accomplished as well as the architectural details, i.e., the number of layers and nodes. What was the computational overhead for training these models?

To clarify this issue, we have added paragraph 3 in the “Model implementation and training” section:

“Model performance is subject to training hyperparameters (e.g., optimizer, learning rate, batch size, number of layers). Hence, an automatic hyperparameter-tuning tool is used, Facebook Ax (https://github.com/facebook/Ax), to search for the optimal hyperparameters for model training. The selected hyperparameters for training the model include a batch size of 16, an SGD optimizer with a learning rate of 0.008 and a momentum of 0.39, and three hidden layers in the MLP. A dropout layer follows each hidden layer in the MLP to prevent overfitting. The number of nodes in each hidden layer is 6836, 5657, and 468, respectively. The dropout rate in each dropout layer is 0.01, 0.07, and 0.67, respectively. The model was trained with four RTX A6000 GPUs. It took about 30 hours to search for the optimal hyperparameters and about an hour to train the model using the optimal hyperparameters.”

3) Consider presenting the data for all significant predictors to justify the choice for inclusion in the model and verify if the important or top features MLP (using explanation methods) uses for prediction are the same as those inferred by the logistic regression.

The AUC of the model using all significant predictors and blastocyst images is 0.75, lower than the AUC (0.77) achieved by the model using logistic regression-selected clinical features and blastocyst images, and the AUC (0.76) achieved by the model using MLP-selected clinical features and blastocyst images. Feature selection reduces the input feature dimensions by removing redundant features and features with limited predictive power, thus improving the model generalization capability.

To follow the reviewer’s suggestion, we used the sequential forward feature selection method to perform MLP-based feature selection. It is explainable in that features are sequentially added to an empty candidate set until the addition of further features does not increase the prediction accuracy. Note that we searched for the optimal MLP parameters (e.g., number of layers, nodes, dropout rate) and training parameters (e.g., optimizer, learning rate) to accurately evaluate the predictive power of each feature candidate.

The MLP-based feature selection method selected 14 clinical features. Compared with the 16 clinical features selected by logistic regression (LR), 12 of the 16 clinical features were selected by both MLP and logistic regression, and the top 9 features are the same. The two clinical features selected by MLP but not by LR include fresh semen (yes/no) and follicle stimulating hormone (FSH) on day 3 after period. The four clinical features selected by LR but not MLP include number of ovarian stimulation cycles, progesterone (P) on HCG day, maternal body mass index (BMI), and free thyroxine (FT4) on day 3 after period. The AUC (0.77) achieved by the model using LR-selected clinical features and blastocyst images and the AUC (0.76) achieved by the model using MLP-selected clinical features and blastocyst images show no significant difference (p-value = 0.95 > 0.05).

Furthermore, the MLP-based feature selection took a few days due to the processes of iterative feature searching/testing and tuning hyperparameters. The LR-based feature selection is much more efficient and only takes a few minutes.

To clarify this issue, we have now added the data regarding the AUC of the model using all significant predictors in and blastocyst images and the MLP-based feature selection results in Figure 3-supplement 1, 2.

4) Please, consider presenting in detail a weighted sampling approach used to tackle the imbalance issue.

To clarify this issue, we have added details regarding the weighted sampling approach in paragraph 2 in the “Model implementation and training” section:

“In the weighted sampling approach, the probability of each item to be selected is determined by its weight, and the weight of each item is assigned by inverse class frequencies. In this way, the weighted sampling approach rebalances the class distributions by oversampling the minority class and under-sampling the majority class.”

5) The code is available only for generating figures 2, 4 reported in the paper. For figure 3, only data is available. Consider presenting this code for reproducibility purposes.

We have provided the source code for generating figure 3. The source code has been submitted with the revised manuscript and can also be accessed at https://github.com/robotVisionHang/LiveBirthPrediction_Data_Code

6) Please, improve the discussion of the potential applications of the proposed model in clinical settings and mention the method of testosterone assessment as a study limitation.

1) The potential application of the proposed model in clinical settings is to improve blastocyst selection. Among the various factors contributing to IVF outcomes, the quality (i.e., developmental potential) of the selected blastocyst for transfer is a major factor determining IVF success. Existing approaches for evaluating and selecting blastocysts are based on manually observing blastocyst morphology grade, which have shown limited predictive power on live birth outcomes of blastocysts (e.g., AUC = 0.58-0.61).

The proposed model achieved a significantly higher accuracy in evaluating the live birth potential of blastocysts (AUC = 0.77) for best blastocyst selection to improve the live birth rate. When the proposed model is applied in clinical practice for blastocyst selection, it takes images of multiple blastocysts of a same patient and patient couple’s clinical features as input, outputs the live birth probability of each blastocyst, and identifies the best blastocyst having the highest live birth probability.

To clarify this issue, we have added following contents on page 13:

“The proposed live birth prediction model improves the evaluation of a blastocyst in terms of its live birth potential for best blastocyst selection from multiple blastocysts of a patient. The next step is to validate the model’s prediction accuracy using prospectively collected data and verify its effectiveness in blastocyst selection via a randomized controlled trial (RCT). Patients enrolled in the RCT will be split into the study group and the control group (1:1 ratio). In the study group, the model selects a top blastocyst having the highest probability of live birth for transfer, and in the control group, embryologists select a top blastocyst based on their routine morphological grading for transfer. Live birth outcomes of both groups will be tracked and compared.”

2) The testosterone (T) included in the clinical feature analysis is total T measuring both free T and bioavailable T.

To clarify this issue, we revised the name of testosterone in Supplementary file 1, which now reads: “Total testosterone on day 3 after period”.

We also mentioned this as a study limitation on page 12:

“Note that in this study, only the total testosterone (T) was analyzed, and free T or bioavailable T was not available for clinical feature analysis (see Supplementary file 1). This may cause potential bias in determining the significance of testosterone as a predictor of live birth.”

Reviewer #1 (Recommendations for the authors):The authors could mention the method of testosterone assessment as a study limitation that may cause a potential bias regarding the estimation of significance of testosterone as a predictor of live birth.

To clarify this issue, we revised the name of testosterone in Supplementary Table Ⅰ, which now reads: “Total testosterone on day 3 after period”.

We also mentioned this as a study limitation on page 12:

“Note that in this study, only the total testosterone (T) was analyzed, and free T or bioavailable T was not available for clinical feature analysis (see Supplementary file 1). This may cause potential bias in determining the significance of testosterone as a predictor of live birth.”

The authors could consider presenting the data for all significant predictors (of example, in the supplemental Figure 3 data) and justify the choice for inclusion in the model. It will better demonstrate the correctness of the selection of predictors

The AUC of the model using all significant predictors and blastocyst images is 0.75, lower than the AUC (0.77) achieved by the model using logistic regression-selected clinical features and blastocyst images, and the AUC (0.76) achieved by the model using MLP-selected clinical features and blastocyst images. Feature selection reduces the input feature dimensions by removing redundant features and features with limited predictive power, thus improving the model generalization capability.

To clarify this issue, we have added the ROC curve comparisons Figure 3-supplement 1:

Reviewer #2 (Recommendations for the authors):I quite enjoyed reading the article. Overall, the motivation and concepts are well defined; however, the manuscript lacks the necessary methodological details, which hindered my ability to fully understand and appreciate the work.The link between CNN and MLP architecture in the final integrated model is unclear. The section "Model architecture" needs to be expanded. It is not clear what "decision-level features" are from CNN or MLP. Are these features from the CNN's fully connected layer? In MLP, are they before the final layer? And how do authors concatenate these features? These details are important to understand final architecture.

The decision-level features are generated from the last fully connected layer in the CNN and the last fully connected layer in the MLP. Each decision-level feature has two variables used to classify the blastocyst into the positive or negative live birth outcome category. The two decision-level features are fused by the adding operation.

To clarify this issue, we have revised the “Model architecture” section, added text in Figure 1 to indicate the last fully connected layer, and provided the code to reproduce the model. The source code has been submitted with the revised manuscript and can also be accessed at https://github.com/robotVisionHang/LiveBirthPrediction_Data_Code.

The revised “Model architecture” section now reads:

“Figure 1 shows the architecture of the live birth prediction model based on multi-modal blastocyst evaluation. It consists of a CNN to process blastocyst images and an MLP to process patient couple's clinical features. Features from the CNN and the MLP are fused; thus, the model can be trained to simultaneously take into account both blastocyst images and patient couple's clinical features for live birth prediction. The last fully connected layer in the CNN and the last fully connected layer in the MLP each output a decision-level feature, which has two variables used to classify the blastocyst into the positive or negative live birth outcome category. The adding operation fuses decision-level features from the CNN and the MLP, and the result of addition is taken as the final output of the overall live birth prediction model.”

How was parameter optimization accomplished? Architectural details, i.e., the number of layers and nodes, are missing. What was the computational overhead for training these models?

To clarify this issue, we have added paragraph 3 in the “Model implementation and training” section:

“Model performance is subject to training hyperparameters (e.g., optimizer, learning rate, batch size, number of layers). Hence, an automatic hyperparameter-tuning tool is used, Facebook Ax (https://github.com/facebook/Ax), to search for the optimal hyperparameters for model training. The selected hyperparameters for training the model include a batch size of 16, an SGD optimizer with a learning rate of 0.008 and a momentum of 0.39, and three hidden layers in the MLP. A dropout layer follows each hidden layer in the MLP to prevent overfitting. The number of nodes in each hidden layer is 6836, 5657, and 468, respectively. The dropout rate in each dropout layer is 0.01, 0.07, and 0.67, respectively. The model was trained with four RTX A6000 GPUs. It took about 30 hours to search for the optimal hyperparameters and about an hour to train the model using the optimal hyperparameters.”

It is difficult to understand the discrepancy of features between CNN with clinical features and CNN without clinical features. Maybe it is because model architecture is not well defined. For the moment, it seems like CNN was trained independently, even in the concatenated version of the model. How can you explain the discrepancy between the activation maps of these two models?

The CNN and the MLP are connected by the two decision-level features. The adding operation fuses decision-level features from the CNN and the MLP, and the addition result is taken as the final output of the overall live birth prediction model. Thus, the model weights of the CNN and the MLP were trained simultaneously.

Another finding of this study, by comparing the heatmaps of the CNN trained without and with the inclusion of patient couple's clinical features, is that the weights of TE-related features increased (see Figure 4). A potential reason may be that TE and the endometrium status-related features (e.g., endometrium thickness and pattern) play critical roles when a blastocyst initiates implantation, and a positive live birth outcome is not possible without the success of this implantation process (Ahlström et al., 2011; Hill et al., 2013; Chen et al., 2014; Bakkensen et al., 2019).

To clarify this issue, we have revised the “Model architecture” section. The explanation of the discrepancy between the heatmaps of the CNN trained without and with clinical features is added on page 12:

“Another finding of this study, by comparing the heatmaps of the CNN trained without and with the inclusion of patient couple's clinical features, is that the weights of TE-related features increased (see Figure 4). A potential reason may be that TE and the endometrium status-related features (e.g., endometrium thickness and pattern) play critical roles when a blastocyst initiates implantation, and a positive live birth outcome is not possible without the success of this implantation process (Ahlström et al., 2011; Hill et al., 2013; Chen et al., 2014; Bakkensen et al., 2019).”

Important features were identified using logistic regression. I do not observe the link presenting these features as important features when the model was built using MLP instead. Could you verify that the important or top features MLP (using explanation methods) uses for prediction are the same as those inferred by the logistic regression?

To follow the reviewer’s suggestion, we used the sequential forward feature selection method to perform MLP-based feature selection. It is explainable in that features are sequentially added to an empty candidate set until the addition of further features does not increase the prediction accuracy. Note that we searched for the optimal MLP parameters (e.g., number of layers, nodes, dropout rate) and training parameters (e.g., optimizer, learning rate) to accurately evaluate the predictive power of each feature candidate.

The MLP-based feature selection method selected 14 clinical features. Compared with the 16 clinical features selected by logistic regression (LR), 12 of the 16 clinical features were selected by both MLP and logistic regression, and the top 9 features are the same. The two clinical features selected by MLP but not by LR include fresh semen (yes/no) and follicle stimulating hormone (FSH) on day 3 after period. The four clinical features selected by LR but not MLP include number of ovarian stimulation cycles, progesterone (P) on HCG day, maternal body mass index (BMI), and free thyroxine (FT4) on day 3 after period. The AUC (0.77) achieved by the model using LR-selected clinical features and blastocyst images and the AUC (0.76) achieved by the model using MLP-selected clinical features and blastocyst images show no significant difference (p-value = 0.95 > 0.05).

Furthermore, the MLP-based feature selection took a few days due to the processes of iterative feature searching/testing and tuning hyperparameters. The LR-based feature selection is much more efficient and only takes a few minutes.

To clarify this issue, we have now added the data regarding the AUC of the model using all significant predictors and blastocyst images and the MLP-based feature selection results in Figure 3-supplement 1, 2:

The imbalance issue was tackled using a weighted sampling approach. This approach needs to be detailed in the main text. And how were train, val, and test partitions built in view of the distribution of minority and majority classes? Did the author verify other approaches that can help resolve this issue?

We have added details regarding the weighted sampling approach in paragraph 2, the “Model implementation and training” section. We also verified the approach of using weighted cross-entropy loss. We found that both approaches can be used to mitigate the prediction bias towards the majority class. The weighted sampling approach performed better and was selected for the final model training.

The stratified random sampling approach was used to ensure that all split datasets (training, validation, and testing datasets) have the same distribution of minority and majority classes.

To clarify this issue, we have revised the following contents in paragraph 2, the “Model implementation and training” section:

“The blastocysts were randomly split into 80%:10%:10% to construct the training, validation, and testing datasets. The stratified random sampling approach was used to ensure that all split datasets have the same distribution of minority and majority classes. Since the ratio of blastocysts with a positive live birth outcome in the dataset is 0.368, to mitigate the model's prediction bias towards the majority category (i.e., the negative live birth outcome), the weighted sampling approach, which can help rebalance the class distributions when sampling from an imbalanced dataset (Feng et al., 2021), was employed for training the model. In the weighted sampling approach, the probability of each item to be selected is determined by its weight, and the weight of each item is assigned by inverse class frequencies. In this way, the weighted sampling approach rebalances the class distributions by oversampling the minority class and under-sampling the majority class. We also verified the approach of using weighted cross-entropy loss, which assigns greater weights to the loss caused by the prediction error of minority classes. Both approaches helped mitigate the prediction bias towards the majority class, and the results showed that the weighted sampling approach outperformed the weighted cross-entropy loss method.”

The code to reproduce the model is missing.

We have provided the source code to reproduce the model. The source code has been submitted with the revised manuscript and can also be accessed at https://github.com/robotVisionHang/LiveBirthPrediction_Data_Code

Discussion regarding implementation in a clinical setting would be informative. How feasible is the model's deployment in a clinic? Maybe you can further elaborate on prospective clinical trial which was mentioned in line # 363.

The potential application of the proposed model in clinical settings is to improve blastocyst selection. Among the various factors contributing to IVF outcomes, the quality (i.e., developmental potential) of the selected blastocyst for transfer is a major factor determining IVF success. Existing approaches for evaluating and selecting blastocysts are based on manually observing blastocyst morphology grade, which have shown limited predictive power on live birth outcomes of blastocysts (e.g., AUC = 0.58-0.61).

The proposed model achieved a significantly higher accuracy in evaluating the live birth potential of blastocysts (AUC = 0.77) for best blastocyst selection to improve the live birth rate. When the proposed model is applied in clinical practice for blastocyst selection, it takes images of multiple blastocysts of a same patient and patient couple’s clinical features as input, outputs the live birth probability of each blastocyst, and identifies the best blastocyst having the highest live birth probability.

To clarify this issue, we have added following contents on page 13:

“The proposed live birth prediction model improves the evaluation of a blastocyst in terms of its live birth potential for best blastocyst selection from multiple blastocysts of a patient. The next step is to validate the model’s prediction accuracy using prospectively collected data and verify its effectiveness in blastocyst selection via a randomized controlled trial (RCT). Patients enrolled in the RCT will be split into the study group and the control group (1:1 ratio). In the study group, the model selects a top blastocyst having the highest probability of live birth for transfer, and in the control group, embryologists select a top blastocyst based on their routine morphological grading for transfer. Live birth outcomes of both groups will be tracked and compared.”